# TVMamba: Towards Efficient Visual Mamba with Ternary Weights and Activations

## Abstract

Visual Mambas built on state space models (SSMs) have recently emerged as powerful vision backbones. To improve efficiency on resource-constrained devices, increasing work explores *quantization* to represent weights and activations at low precision. However, state-of-the-art methods typically reach ternary weights while activations remain at 8 bits or higher. We propose *TVMamba*, the first to achieve *ternary weights and activations* to our knowledge. Specifically, our analyses indicate that uneven channel distributions make ternary activations difficult, causing unstable optimization and amplified spectral distortions. To address this, TVMamba employs two components: (1) a *quinary-to-ternary* staged codebook that trains with five-level activations for stability and collapses to ternary at deployment and (2) a lightweight *quantization-aware frequency routing* module that preserves high-frequency detail while maintaining the SSM core's low-pass strength. Empirically, on two mainstream Visual Mamba backbones (VMamba, Vim), our method delivers competitive accuracy. Wall-clock measurements across devices show `MatMul` at various sizes accelerated by $17\times$–$87\times$ via joint weight–activation ternarization.

## 1 Introduction

Recent efforts apply Mamba (Gu et al., 2021a;b; Gu & Dao, 2023) in the visual domain, demonstrating competitive performance across multiple benchmarks (Huang et al., 2024; Liu et al., 2024; Zhu et al., 2024; Xiao et al., 2025). For on-device deployment of large-scale architectures, quantization provides an efficient compression strategy by representing weights and activations at low precision (Lin et al., 2024; Frantar et al., 2023; He et al., 2023). Compared with well-established quantization practices for Transformer-based vision models (Dettmers et al., 2022; Le & Li, 2023; Xiao et al., 2023; Gao et al., 2025), the Mamba family remains underexplored.

Existing Mamba quantization largely adopts post-training quantization (PTQ), typically setting both weights and activations to 8 bits to preserve accuracy (Chiang et al., 2025; Xu et al., 2025). Only a handful of quantization-aware training (QAT) methods push weights to binary or ternary (Tang et al., 2024; Yu et al., 2025). However, activations generally remain at 8-bit or higher, limiting the potential for efficient inference.

In light of this, we develop TVMamba, a deployment-efficient framework that jointly ternarizes weights and activations. We profile two representative Visual Mamba backbones, VMamba Liu et al. (2024) and Vision Mamba (Vim) Zhu et al. (2024), and report the parameter breakdowns in Table 1. The analysis indicates that linear projection layers dominate the model size, thus prioritizing their ternarization yields the largest gains in storage reduction and inference speed.

When initialized from pretrained full-precision checkpoints, constraining activations to ternary levels in QAT tends to destabilize optimization and risks collapse without careful design. Even with successful convergence, accuracy suffers a significant drop. Analysis of full-precision activations reveals heavy-tailed, anisotropic channel distributions with outliers concentrated in a small subset of channels. Under quantization, the SSM pathway's intrinsic low-pass bias is accentuated, disproportionately eroding high-frequency structure such as edges and fine textures.

To address these issues, we introduce a quinary-to-ternary activation quantization scheme that begins with a learnable, per-channel quinary quantizer to buffer heavy-tailed responses, and then anneals to

Table 1: Module-wise breakdown of trainable parameters (%) in VMamba (Liu et al., 2024) and Vim (Zhu et al., 2024) backbones. Linear projection layers account for the majority.

| Model | Embedding | LayerNorm | LinearProjection | SelectiveScan | Convolution | Others |
|-------|-----------|-----------|------------------|---------------|-------------|--------|
| VMamba-S | 0.09% | 0.13% | **84.89%** | 7.57% | 0.28% | 6.95% |
| VMamba-B | 0.09% | 0.10% | **85.15%** | 7.47% | 0.21% | 7.00% |
| Vim-S | 1.14% | 0.04% | **83.80%** | 14.00% | 0.71% | – |
| Vim-B | 0.77% | 0.02% | **87.82%** | 11.02% | 0.38% | – |

true ternary under a small compute budget. To further stabilize ternary training, we add a Channel-wise Tail-Risk Loss (CTRL) to the task loss, which focuses learning on rare, high-magnitude errors and steers the per-channel scales to absorb outliers without harming the bulk activations. We control the number of five-level channels via a bits-to-accuracy trade-off, retaining more when latency and energy budgets permit, and annealing to zero under tight constraints. At deployment, each five-level activation decomposes exactly into the sum of two ternary matrix multiplications, enabling reuse of standard ternary kernels.

Despite the stabilized activation distribution afforded by our quinary-to-ternary quantizer, fine-scale cues remain brittle under ternary constraints. We therefore insert a lightweight, quantization-aware frequency router (QAFR) which applies a learnable low-high decomposition before the Selective Scan core of the SSM block. Then we compute channel-wise mixing coefficients conditioned on content and quantization signals. These coefficients route low-frequency components to the native SSM and direct high-frequency components to a light-weight enhancer branch. With a high-frequency error spectrum distillation term, QAFR incurs negligible overhead and complements quinary-to-ternary annealing to deliver more stable accuracy under fully ternary deployment.

The major contributions of this paper are summarized as follows:

- We introduce TVMamba, to our knowledge the first framework to ternarize both weights and activations in Visual Mamba. A staged quinary-to-ternary activation quantizer, coupled with a Channel-wise Tail-Risk (CTRL) loss, stabilizes ternary training.

- To mitigate quantization-amplified spectral distortions, we introduce a Quantization-Aware Frequency Routing (QAFR) module to perform a learnable low-high decomposition. With a high-frequency error-spectrum distillation objective, QAFR adds negligible overhead while preserving high-frequency details under ternary constraints.

- We conduct comprehensive experiments on two Visual Mamba backbones across classification, detection, and segmentation, demonstrating competitive accuracy with end-to-end latency and energy reductions consistent with bit-ops savings.

## 2 RELATED WORK

**Visual state space models (SSMs).** Recently, State Space Models (SSMs) (Fu et al., 2023; Lieber et al., 2025) have emerged as compelling alternative to Vision Transformers (ViTs) (Vaswani et al., 2017; Dosovitskiy et al., 2021). Modern visual mamba backbones (Liu et al., 2024; Zhu et al., 2024; Shaker et al., 2025; Huang et al., 2024; Xiao et al., 2025) are built on discretized selective state space models equipped with parallel scans for linear-time sequence computation. VMamba Liu et al. (2024) adapts Mamba to 2D vision through the 2D-Selective-Scan (SS2D) where image patches are traversed along four complementary cross-scan routes. Each route is processed by a selective SSM block in parallel, thus preserving global receptive fields. In contrast, Vision Mamba (Vim) Zhu et al. (2024) employs bidirectional scans on a 1D token sequence augmented by positional embeddings and a CLS token. LocalVMamba Huang et al. (2024) addresses the challenge of capturing detailed local information by introducing a scanning methodology within distinct windows (inspired from Swin Transformer Liu et al. (2021)), coupled with dynamic scanning directions across network layers. EfficientVMamba Pei et al. (2025) integrates atrous-based selective scanning and dual-pathway modules for efficient global and local feature extraction, achieving competitive results with reduced computational complexity. These models have been applied widely for various vision tasks (Gong et al., 2025; Gu et al., 2021b; Chen et al., 2024), demonstrating the effectiveness of SSMs (Gu et al., 2021a; Mehta et al., 2022), and in particular Mamba (Gu & Dao, 2023), in the visual domain.

**Mamba quantization.** Quantization methods are often grouped by whether additional training is used, namely quantization-aware training (QAT) (Esser et al., 2019; Lee et al., 2023; Shin et al., 2023; Zhou et al., 2016) and post training quantization (PTQ) (Banner et al., 2019; Kim & Park, 2024; Li et al., 2022; Nagel et al., 2019; So et al., 2023; Wei et al., 2023). The linear recurrence in state space models creates distinctive difficulties for quantization, especially for activations, since rare extreme values are hard to represent. Current PTQ work on SSMs (Chiang et al., 2025; Xu et al., 2025; Ramachandran et al., 2025) mainly relies on rotation based preprocessing. For example, Quamba (Chiang et al., 2025) applies static 8 bit per tensor quantization with a Hadamard transform to smooth activation distributions, and MambaQuant (Xu et al., 2025) employs KLT based and Smooth Fused rotations with a similar goal. These approaches largely remain at 8 bit precision and are mainly demonstrated on language models. By comparison, QAT, through optimization during training, offers a practical route to very low bit settings. Slender Mamba (Yu et al., 2025) ternarizes Mamba 2 (Dao & Gu, 2024) and pretrains it from scratch on 150B tokens, and Bi-Mamba (Tang et al., 2024) further reduces weights to a single bit while retaining competitive accuracy. In both directions, however, activations are typically kept in full precision or 8 bit, which limits achievable inference speedups and memory savings and leaves considerable efficiency headroom.

## 3 PRELIMINARIES

### 3.1 STATE SPACE MODELS

Inspired by continuous-time systems, conventional state space models (S4) (Gu et al., 2021a) capture sequence context with linear-time complexity. Following a linear time-invariant (LTI) state-space formulation, these models map an input signal $x(t) \in \mathbb{R}$ to an output response $y(t) \in \mathbb{R}$ through a hidden state $h(t) \in \mathbb{R}^N$:

$$\dot{h}(t) = \mathbf{A}h(t) + \mathbf{B}x(t), \qquad y(t) = \mathbf{C}h(t) + \mathbf{D}x(t) \tag{1}$$

where $\mathbf{A} \in \mathbb{R}^{N \times N}$ is the transition state matrix, $\mathbf{B} \in \mathbb{R}^{N \times 1}$, $\mathbf{C} \in \mathbb{R}^{1 \times N}$ are the projection parameters, and $\mathbf{D} \in \mathbb{R}^1$ denotes a learnable direct feed-through term.

For compatibility with gradient-based training and parallelism, the continuous SSM is discretized via a time-scale $\mathbf{\Delta}$, replacing $\mathbf{A}, \mathbf{B}$ with their discrete forms: $\overline{\mathbf{A}} = \exp(\mathbf{\Delta}\mathbf{A})$ and $\overline{\mathbf{B}} = (\mathbf{\Delta}\mathbf{A})^{-1}(\exp(\mathbf{A}) - I)\mathbf{\Delta}\mathbf{B}$. Given the discretized parameters, the system admits the discrete-time recurrence:

$$h_t = \overline{\mathbf{A}}h_{t-1} + \overline{\mathbf{B}}x_t, \qquad y_t = \mathbf{C}\,h_t + \mathbf{D}\,x_t. \tag{2}$$

To enable parallel computation during training, a structured convolutional kernel $\overline{\mathbf{K}}$ is introduced to compute the output as follows:

$$\overline{\mathbf{K}} = (\mathbf{C}\overline{\mathbf{B}}, \ \mathbf{C}\overline{\mathbf{A}}\overline{\mathbf{B}}, \ \ldots, \ \mathbf{C}\overline{\mathbf{A}}^{L-1}\overline{\mathbf{B}}), \qquad y = x * \overline{\mathbf{K}}, \tag{3}$$

where $L$ is the length of the input sequence $x$.

To further improve content-aware sequence modeling, Selective State Space models (S6) (Gu & Dao, 2023) parameterize the state-space parameters as functions of the input, enabling the model to selectively propagate or forget information along the sequence.

### 3.2 TERNARY QUANTIZATION

Ternary quantization has been explored across multiple architectures, *e.g.*, convolutional neural networks (Zhu et al., 2017; Chen et al., 2021) and Transformers (Xu et al., 2022; Kaushal et al., 2025; Grainge et al., 2025). A straightforward application to weights uses a single global (per-tensor) scaling factor for ternarization. Concretely, for a full-precision weight matrix $\mathbf{W} \in \mathbb{R}^{d_{in} \times d_{out}}$, the global scale used for ternarization is computed as

$$s_w = \max\left(\frac{1}{d_{\text{in}}\,d_{\text{out}}} \sum_{i,j} \mathbf{W}_{ij}|, \ \varepsilon\right), \tag{4}$$

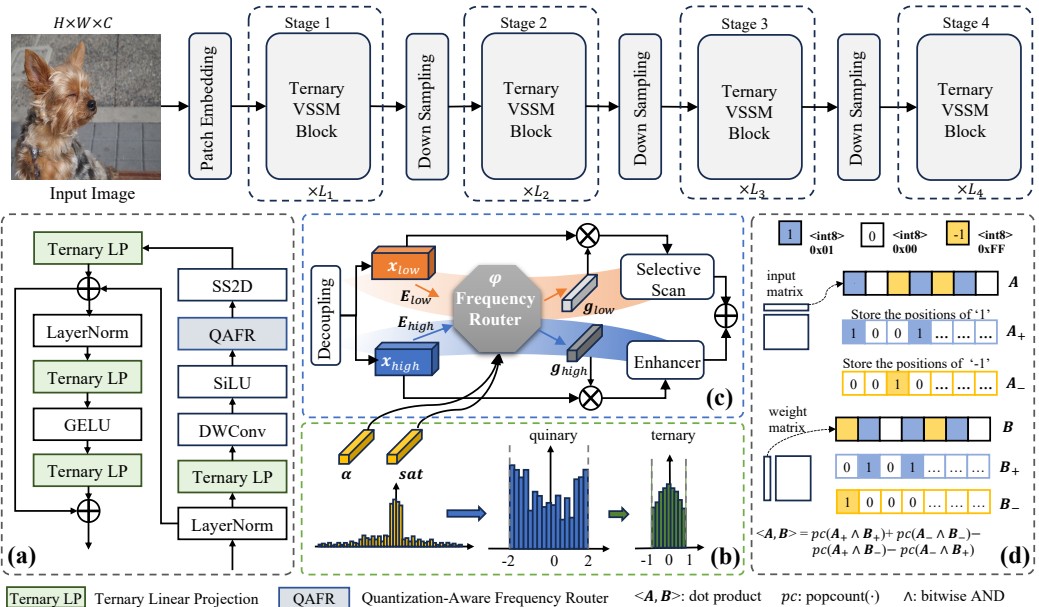

Figure 1: Overview of **TVMamba**. **(a) Ternary SSM block** for visual Mamba backbones: weights and activations in all linear projection layers are ternarized. **(b) Quinary-to-Ternary (Q2T) activation quantizer:** activations are trained with a quinary codebook and collapsed to ternary for deployment. **(c) Quantization-aware frequency routing (QAFR):** features are decomposed into low/high bands. Low frequencies are routed to SS2D, while high-frequency detail is preserved via an enhancement branch. **(d) Ternary bit-wise computation:** matrix multiplication is implemented with bit-wise encodings and popcount-style accumulators.

where $\epsilon > 0$ is a small constant to avoid division to zero. A ternary proxy $\widetilde{\mathbf{W}} \in \{-1, 0, 1\}^{d_{in} \times d_{out}}$ is then obtained via saturated rounding:

$$\widetilde{\mathbf{W}} = \text{clip}\big(\text{round}(\frac{\mathbf{W}}{s_w}), -1, 1\big), \qquad \mathbf{W} \approx s_w \widetilde{\mathbf{W}}, \tag{5}$$

where $\text{round}(\cdot)$ maps each entry to the nearest integer and $\text{clip}(x, -1, 1)$ saturates values to $[-1, 1]$.

Pre-activations are normalized prior to quantization by inserting a token-wise normalization step, after which activations are ternarized with learnable per-channel parameters. A mean-free RMS normalization (Zhang & Sennrich, 2019) is employed to stabilize scale and preserve variance under low-bit mapping.

To ensure trainability and stable convergence, activations are commonly quantization using learnable parameters. With a per-channel learnable scale $\boldsymbol{\alpha}$ and a per-channel learnable threshold ratio $\mathbf{k}$, the threshold can be defined as $\boldsymbol{\Delta} = \mathbf{k}\boldsymbol{\alpha}$. Then activations are quantized by

$$\mathbf{q} = \boldsymbol{\alpha} \, \text{sign}(\mathbf{x}) \, \mathbf{1}(|\mathbf{x}| \geq \boldsymbol{\Delta}), \qquad q \in \{-\boldsymbol{\alpha}, 0, \boldsymbol{\alpha}\}, \tag{6}$$

where $\text{sign}(\cdot)$ is the sign function, and $\mathbf{1}$ is the indicator. During quantization-aware training, the non-differentiable operations are handled with a straight-through estimator (Bengio et al., 2013).

## 4 METHOD

In this section, we systematically introduce our approach. We first analyze the challenges of activation ternarization in Section 4.1 and then describe our TVMamba in Section 4.2. Figure 1 presents the overall framework of the proposed TVMamba model. All linear projection layers in visual SSM blocks are replaced by ternary counterparts, coupled with the proposed quinary-to-ternary (Q2T) activation quantizer described in Section 4.2.1. A quantization-aware frequency routing module is inserted before the Selective Scan, with details in Section 4.2.2.

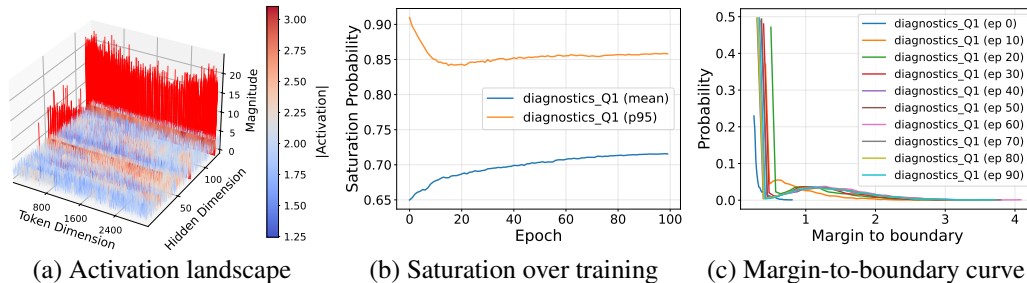

(a) Activation landscape  (b) Saturation over training  (c) Margin-to-boundary curve

Figure 2: **(a)** 3D visualization of *In_Proj* activations across tokens and channels. Tall red spikes highlight outlier responses. **(b)** Saturation probability over training (per-channel mean and 95th percentile), revealing early over-saturation and its slow drift. **(c)** Margin-to-boundary curve for unsaturated samples, showing heavy mass near the ternary thresholds with a long tail.

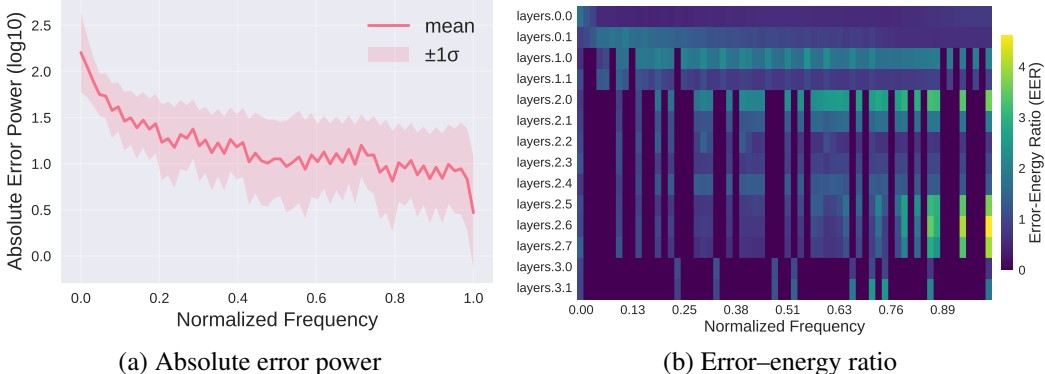

(a) Absolute error power  (b) Error–energy ratio

Figure 3: **(a)** Absolute error spectrum of SSM outputs (ternary *vs.* full-precision). **(b)** Error–energy ratio heatmap across layers and normalized frequency.

## 4.1 WHY TERNARY QUANTIZATION IS CHALLENGING?

We first fine-tune a full-precision, pretrained VMamba with strict ternary quantization on its linear projections, but training collapsed early. This motivate us to instrument the model and examined the In_Proj activation distributions in Figure 2(a). The visualization shows a strongly skewed channel distribution with extreme responses concentrated in a few hot channels, which distorts per-channel scale and threshold estimates. To assess how often activations hit the boundaries, Figure 2(b) reports per-channel saturation—the fraction of tokens whose normalized magnitude exceeds the learned threshold. The mean increases while the 95th percentile remains high, indicating that many samples operate at or beyond the limits. A complementary view in Figure 2(c) plots the margin density for non-saturated samples and concentrates near zero with a shallow right tail, showing that most activations lie close to the ternary thresholds.

These findings indicate an optimization pathology: early ternarization compresses dynamic range, pushes gradients toward decision boundaries, and amplifies inter-channel imbalance. Motivated by this, we adopt a quinary-to-ternary activation quantizer that temporarily expands the effective range and stabilizes per-channel scale estimation. This progressive strategy mitigates the imbalance that undermines naive ternarization, yielding a smoother, more trainable quantization trajectory.

To further analyze how ternary linear layers affect the SSM branch, we compare the SSM outputs of the ternary model with a full-precision teacher in the frequency domain. Figure 3(a) plots the absolute error power against frequency and shows that errors concentrate at low frequencies and decay as frequency increases. Figure 3(b) presents the error–energy ratio across layers and frequencies and reveals large relative errors in the high-frequency bands, most notably in deeper layers. These results indicate that quantization biases low-frequency amplitudes and destabilizes high-frequency details. We therefore introduce a frequency-aware routing module that sends low-frequency content through the original SSM and handles high-frequency content with a light-weight enhancer, which reduces low-frequency bias and suppresses high-frequency artifacts.

## 4.2 THE PROPOSED TVMAMBA

### 4.2.1 QUINARY-TO-TERNARY ACTIVATION QUANTIZER

We follow the quantization setup introduced in Section 3.2 for linear operators. For activation quantization, we deliberately remove the learnable threshold and retain only a per-channel learnable scale, which simplifies optimizations and makes the annealing from quinary to strick ternary well-conditioned. We denote the post-normalization activation as $\overline{\mathbf{x}} \in \mathbb{R}^{N \times C}$. With a learnable per-channel scale $\boldsymbol{\alpha}$ and an integer radius $Q \in \{2, 1\}$, the forward quantization mapping is

$$\mathbf{q}_Q = \text{clip}\big(\text{round}(\frac{\overline{\mathbf{x}}}{\boldsymbol{\alpha}}), -Q, Q\big), \qquad y = \boldsymbol{\alpha}\mathbf{q}_Q. \tag{7}$$

During the first one-third of training epochs, all channels use quinary codes ($Q = 2$). For the remainder of training, we invoke the online bits-to-accuracy allocation every $T$ steps to update each channel's code radius, gradually annealing to ternary ($Q = 1$).

To explicitly suppress outlier-induced errors within each channel while keeping the implementation lightweight, we add a local tail-risk regularizer at the quantized layer. For channel $c$, let $\mathbf{e}_{c,i} = \|\hat{\mathbf{x}}_{c,i} - \mathbf{x}_{c,i}\|_2^2$ be the per-example reconstruction error between quantized and full-precision activations. We define the CTRL as the top-$k\%$ tail mean:

$$\mathcal{L}_{\text{CTRL}} = \sum_c \text{Top-}k\text{-Mean}\big(\{\, \mathbf{e}_{c,i} \,\}_{i=1}^N\big). \tag{8}$$

For each candidate channel $c$, we obtain the risk reduction $\Delta R_c$ by toggling $Q_c$ between ternary and quinary while keeping all other channels unchanged. To avoid $C$ separate forward passes, we estimate the per-channel benefit in a single backward pass via a first-order influence. Let $\mathbf{y} = \mathbf{W}\mathbf{x}$ be the linear output, $g^{\text{in}}$ the input-side gradient (available from backpropagation), and $\delta\mathbf{x}_c = \mathbf{x}_c^{(5)} - \mathbf{x}_c^{(3)}$ the activation difference at the quantized layer when toggling only channel $c$. We score channels by

$$\Delta R_c \approx \text{Top-}k\text{-Mean}\big(g_c^{\text{in}} \odot \delta x_c\big), \tag{9}$$

which yields all $\{\Delta R_c\}_{c=1}^C$ without any extra forward passes.

We adopt a unit-cost model and set the per-channel compute increment to $\Delta B_c = 1$, i.e., keeping a channel quinary counts as one budget unit. With a global budget $B_{\text{max}}(t)$ (the maximum number of quinary channels allowed at step $t$), we solve a 0–1 knapsack via a greedy ratio rule that reduces to sorting by $\Delta R_c$:

$$\max_{m_c \in \{0,1\}} \sum_c m_c \Delta R_c \quad \text{s.t.} \quad \sum_c m_c \leq B_{\text{max}}(t). \tag{10}$$

Every $T$ steps we pick the top-$B_{\text{max}}(t)$ channels by $\Delta R_c$ and decay $B_{\text{max}}(t)$ over epochs to anneal from quinary to ternary. Details of this procedure can be found in the Appendix A.3.

### 4.2.2 QUANTIZATION-AWARE FREQUENCY ROUTING

Low-bit training of linear projections compresses the dominant compute of visual Mamba blocks but accentuates their intrinsic low-pass tendency, suppressing genuine high-frequency (HF) structures and occasionally introducing spurious HF residuals. To mitigate these effects without perturbing the recurrent core, we augment each SSM block with a *quantization-aware frequency routing* mechanism that preserves the SSM's advantage on low-frequency (LF) content while delegating HF content to a quantization-friendly bypass. The module is lightweight, differentiable end-to-end, and fully compatible with the quinary-to-ternary activations quantizers described in Section 4.2.1.

Given an intermediate feature map $\mathbf{z} \in \mathbb{R}^{B \times C \times H \times W}$ before the core *Selective Scan*, we obtain a spectral split by a depthwise separable smoothing operator $\mathcal{L}$ whose kernel is non-negative and sums to one:

$$\mathbf{x}_{\text{low}} = \mathcal{L}(\mathbf{z}), \qquad \mathbf{x}_{\text{high}} = \mathbf{z} - \mathbf{x}_{\text{low}}. \tag{11}$$

To decide the processing path in a channel-wise manner, we use a two-layer perceptron $\varphi$ to produce logits that are normalized by a softmax over the two branches, yielding per-channel routing weights

$$[\mathbf{g}_{\text{low}}, \mathbf{g}_{\text{high}}] = \text{softmax}\big(\varphi(\mathbf{E}_{\text{low}}, \mathbf{E}_{\text{high}}, \boldsymbol{\alpha}, \mathbf{sat})\big), \tag{12}$$

Table 2: ImageNet-1K Top-1 accuracy (%) and model storage (MB) on VMamba and Vim backbones at Tiny/Small/Base (T/S/B) scales. Slender-Mamba* replaces 8-bit activations with ternary in VMamba/Vim, following the original Slender-Mamba setup. **TVMamba-F** retains five-level activations during training, while **TVMamba-T** deploys strict ternary activations. Storage indicates the on-disk model size under different precision formats.

| Model | Method | Top-1 (%) | Storage (MB) | Model | Top-1 (%) | Storage (MB) |
|-------|--------|-----------|--------------|-------|-----------|--------------|
| VMamba-T | FP16 | 82.6 | 115.8 | Vim-T | 76.1 | 27.3 |
| | Slender-Mamba* | 44.5 | 25.2 | | 37.7 | 7.6 |
| | LSQ-Ternary | 63.2 | 25.4 | | 57.9 | 7.6 |
| | **TVMamba-F (ours)** | **79.6** | **26.6** | | **73.8** | **7.7** |
| | **TVMamba-T (ours)** | **78.4** | **26.6** | | **72.6** | **7.7** |
| VMamba-S | FP16 | 83.6 | 191.3 | Vim-S | 80.5 | 98.4 |
| | Slender-Mamba* | 45.1 | 39.2 | | 42.1 | 21.1 |
| | LSQ-Ternary | 66.1 | 39.5 | | 62.3 | 21.2 |
| | **TVMamba-F (ours)** | **81.8** | **39.8** | | **78.2** | **21.3** |
| | **TVMamba-T (ours)** | **80.6** | **39.8** | | **77.0** | **21.3** |
| VMamba-B | FP16 | 83.9 | 337.8 | Vim-B | 81.9 | 372.3 |
| | Slender-Mamba* | 45.3 | 68.2 | | 43.5 | 65.8 |
| | LSQ-Ternary | 66.2 | 68.5 | | 63.7 | 66.0 |
| | **TVMamba-F (ours)** | **81.8** | **68.8** | | **79.6** | **66.3** |
| | **TVMamba-T (ours)** | **80.7** | **68.8** | | **78.4** | **66.3** |

where $\mathbf{E}_{\text{low}}, \mathbf{E}_{\text{high}} \in \mathbb{R}^C$ are channel-wise spectral-energy descriptors aggregated over spatial axes, $\boldsymbol{\alpha} \in \mathbb{R}^C$ collects the per-channel scales from the quinary-to-ternary (Q2T) quantizer, and $\text{sat} = \Pr(|z/\alpha| > Q)$ is the per-channel saturation rate computed with detached statistics. $Q \in \{2, 1\}$ denotes the current Q2T code radius at the input projection of the same SSM block.

Then the LF content $y_{\text{low}} = g_{\text{low}} \odot x_{\text{low}}$ is processed by the original recurrent core in higher precision and modulated by the router, whereas the HF content is handled by:

$$\mathbf{y}_{\text{high}} = \mathcal{E}(\mathbf{g}_{\text{high}} \odot \mathbf{x}_{\text{high}}), \qquad \mathcal{E}(\cdot) = \mathbf{W}_2 \, \sigma(\mathbf{W}_1(\cdot)), \tag{13}$$

where $\mathbf{W}_1, \mathbf{W}_2$ are depthwise-separable or $1\times1$ convolutions and $\sigma$ is a pointwise nonlinearity.

To enforce spectral fidelity and align optimization with deployment cost, we couple the task loss with an auxiliary term which penalizes quantization-induced HF distortion via a fixed high-pass operator $\mathcal{H}$:

$$\mathcal{L}_{\text{QHF}} = \left\| \mathcal{H}(\mathbf{y} - \mathbf{y}^{\text{dq}}) - \mathcal{H}(\mathbf{y}^T - (\mathbf{y}^T)^{\text{dq}}) \right\|_2^2, \tag{14}$$

where $\mathbf{y}^T$ is the full-precision teacher output of the same block and the superscript $\text{dq}$ denotes de-quantized signals.

In summary, the proposed routing preserves the SSM's low-frequency competence while delegating fragile high-frequency regions to a sparse, ternary bypass that maps cleanly to bit-wise kernels, yielding consistent gains under ternary weights and activations and predictable latency improvements on edge hardware.

## 5 EXPERIMENTS

### 5.1 EXPERIMENTAL SETTINGS

In order to demonstrate the superiority of our method, we conduct comprehensive experiments across various computer vision tasks, including image classification, object detection and instance segmentation on ImageNet-1k (Russakovsky et al., 2015) and MSCOCO 2017 datasets (Lin et al., 2014). In all experiments, models are fine-tuned for 100 epochs from their corresponding pretrained full-precision checkpoints. Unless otherwise noted, architectural and training settings follow the full-precision counterparts Liu et al. (2024); Zhu et al. (2024), the only change is the learning-rate schedule. We adopt a three-stage schedule: the first third of training uses 0.1× the base learning rate, the middle third uses 0.5×, and the final third uses the base rate. During the first third, all channels remain in the quinary domain, thereafter the quinary-to-ternary annealing (Q2T) is activated. All experiments are conducted on 8× NVIDIA A800 GPUs.

## 5.2 RESULTS ON IMAGE CLASSIFICATION

We evaluate our proposed **TVMamba** using two visual mamba backbones Vmamba (Liu et al., 2024) and Vision Mamba (Vim) (Zhu et al., 2024). Table 2 reports ImageNet-1K Top-1 accuracy and model storage across three different scales. For comparison, we include two quantization baselines. Slender-Mamba* adapts the Slender-Mamba (Yu et al., 2025) quantization recipe to the VMamba and Vim backbones by replacing the original 8-bit activation quantizer with a ternary scheme, while keeping all other settings unchanged. In addition, we implement an LSQ-style learned quantizer (Bhalgat et al., 2020) for activations with per-channel learnable step size and threshold as described in Section 3.2. For our proposed method, we consider two deployment variants: **TVMamba-T** enforces strict ternary activations at inference, while **TVMamba-F** retains the stabilized five-level representation and realizes it as the sum of two ternary matrix multiplications.

From Table 2, we observe that **TVMamba-T** reduces the Top-1 gap to FP16 to 3.2% while achieving 4.35× storage compression on VMamba model, whereas, on Vim it attains a similarly 3.5% gap with 3.55× compression. Retaining the five-level representation at inference, **TVMamba-F** further narrows the FP16 gap on both VMamba and Vim at essentially identical storage, providing a practical accuracy–bit-ops trade-off without inflating parameters. Slender-Mamba*, which employs a fixed activation scaling factor within a binary SSM, consistently shows a pronounced drop in Top-1 accuracy relative to FP16 across both VMamba and Vim backbones and all model scales. Compared with LSQ-Ternary, **TVMamba-T** yields a consistent 14.72% improvement averaged over all six backbones, and **TVMamba-F** reaches 15.90%. These results validate the effectiveness of **TVMamba**, showing that fully ternarized Visual Mamba backbones maintain competitive accuracy while providing consistent compression.

Table 3: Comparison on object detection and semantic segmentation on MSCOCO 2017 dataset.

| Model | Method | $AP^{box}$ | $AP^{mask}$ |
|---|---|---|---|
| VMamba-T | FP16 | 47.3 | 42.7 |
| | Slender-Mamba* | 0.2 | 0.2 |
| | LSQ-Ternary | 22.1 | 20.6 |
| | **TVMamba-F (ours)** | **43.4** | **39.2** |
| | **TVMamba-T (ours)** | **42.1** | **38.0** |
| VMamba-S | FP16 | 48.7 | 43.7 |
| | Slender-Mamba* | 2.1 | 1.2 |
| | LSQ-Ternary | 23.2 | 21.7 |
| | **TVMamba-F (ours)** | **44.5** | **40.4** |
| | **TVMamba-T (ours)** | **43.1** | **37.1** |
| VMamba-B | FP16 | 49.2 | 44.1 |
| | Slender-Mamba* | 2.3 | 1.3 |
| | LSQ-Ternary | 24.4 | 23.0 |
| | **TVMamba-F (ours)** | **45.1** | **40.9** |
| | **TVMamba-T (ours)** | **44.7** | **40.3** |

## 5.3 RESULTS ON OBJECT DETECTION AND SEGMENTATION

We evaluate transfer to dense prediction by fine-tuning on COCO for object detection and instance segmentation using VMamba and Vim backbones at Tiny/Small/Base scales in Table 3. Starting from the LSQ-Ternary baseline, **TVMamba-T** consistently improves both $AP^{box}$ and $AP^{mask}$ across all scales, indicating that the staged quinary-to-ternary activation schedule stabilizes heavy-tailed channels under the stricter optimization regime of dense tasks. In contrast, the **Slender-Mamba*** variant exhibits pronounced degradation on both $AP^{box}$ and $AP^{mask}$. Overall, the dense-task results corroborate our conclusions from classification that combining quinary-to-ternary annealing with frequency-aware routing enables fully ternary Visual Mamba backbones to retain strong performance while delivering predictable compression.

## 5.4 ABLATION STUDY

To quantify the contribution of each component utilized in our **TVMamba**, we perform an ablation starting from the LSQ-Ternary baseline ("baseline" in Table 4. Introducing the quinary-to-ternary quantizer (Q2T) yields the dominant gain, improving Top-1 from 63.2% to 77.5% on VMamba-T (+14.3%) and from 57.9% to 71.6% on Vim-T (+13.7%). This jump corroborates our hypothesis that staging activations through a five-level buffer stabilizes heavy-tailed, anisotropic channels. Adding the quantization-aware frequency routing (QAFR) module brings a further, consistent improvement, indicating that the learnable low–high decomposition and channel-wise mixing protect high-frequency cues without weakening the recurrent SSM core. Finally, incorporating the Q-HF

Table 4: Ablation study on ImageNet top-1 accuracy. "Q2T": quinary-to-ternary quantizer, "QAFR": quantization-aware frequency routing, "Q-HF": high-frequency consistency loss.

| Model | Method | Top-1 | Model | Top-1 |
|---|---|---|---|---|
| VMamba-T | baseline | 63.2 | Vim-T | 57.9 |
| | +Q2T | 77.5 | | 71.6 |
| | +QAFR | 78.0 | | 72.1 |
| | +Q-HF | **78.4** | | **72.6** |

Table 5: Estimated relative energy comparison between floating-point (FP) and bitwise compute across matrix sizes (normalized to FP=1.0, lower is better).

| Matrix size | FP | Bit-wise | Reduction |
|---|---|---|---|
| $2^9$ | 1.00 | 0.1786 | **5.6×** |
| $2^{10}$ | 1.00 | 0.1205 | **8.3×** |
| $2^{11}$ | 1.00 | 0.0730 | **13.7×** |
| $2^{12}$ | 1.00 | 0.0413 | **24.2×** |

Table 6: Matrix multiplication latency (ms) across different devices.

| size | Intel i5-12500H (PC) | | | Intel Xeon 8457C (server) | | |
|---|---|---|---|---|---|---|
| | full precision | ternary | speedup | full precision | ternary | speedup |
| 512 | $4.496 \times 10^2$ | $2.614 \times 10^1$ | **17.2** | $8.829 \times 10^2$ | $4.312 \times 10^1$ | **20.5** |
| 1024 | $5.403 \times 10^3$ | $1.845 \times 10^2$ | **29.3** | $7.132 \times 10^3$ | $3.193 \times 10^2$ | **22.3** |
| 2048 | $1.063 \times 10^5$ | $2.226 \times 10^3$ | **47.7** | $5.982 \times 10^4$ | $2.329 \times 10^3$ | **25.7** |
| 4096 | $8.670 \times 10^5$ | $1.001 \times 10^4$ | **86.6** | $5.533 \times 10^5$ | $1.786 \times 10^4$ | **31.0** |
| size | ARM Cortex-A78AE (Jetson AGX Orin) | | | ARM Cortex-A76 (Raspberry Pi5) | | |
| | full precision | ternary | speedup | full precision | ternary | speedup |
| 512 | $1.502 \times 10^3$ | $5.958 \times 10^1$ | **25.2** | $1.473 \times 10^3$ | $5.827 \times 10^1$ | **25.3** |
| 1024 | $1.426 \times 10^4$ | $3.968 \times 10^2$ | **35.9** | $1.744 \times 10^4$ | $3.835 \times 10^2$ | **45.5** |
| 2048 | $1.181 \times 10^5$ | $2.825 \times 10^3$ | **41.8** | $1.579 \times 10^5$ | $2.751 \times 10^3$ | **57.4** |
| 4096 | $1.020 \times 10^6$ | $2.088 \times 10^4$ | **48.8** | $1.400 \times 10^6$ | $2.074 \times 10^4$ | **67.5** |

loss provides an additional, smaller but stable boost , suggesting that explicitly aligning the high-frequency error spectrum complements QAFR's routing with improved spectral fidelity.

## 5.5 EFFICIENCY ANALYSES

During model inference, `MatMul` dominates both runtime and energy due to its heavy floating-point compute and frequent memory access. Our proposed TVMamba ternarizes both weights and activations, thus enabling to convert most floating-point operations into inexpensive bitwise operations. We simulate `MatMul` with full precision and ternary bitwise execution across PC/server CPUs and ARM-based edge SoCs over a range of matrix sizes in Table 6. In addition, using Horowitz's energy data (Horowitz, 2014), Table 5 presents the energy of full-precision `MatMul` versus bitwise matrix multiplication for a linear layer. These results show that our approach effectively exploits bitwise computation, yielding substantial advantages in both matrix-multiplication speed and energy.

## 6 CONCLUSION

In this paper, we introduce TVMamba, a quantization-aware training framework that, to our knowledge, is the first to jointly ternarize both weights and activations in visual Mamba models. We begin with a comprehensive analysis of ternary activation quantization, visualizing activation distributions and comparing frequency-domain errors against the full-precision model. The analyses show that uneven channel statistics hinder ternary activations, causing optimization instability and amplified spectral distortions. To address this, our method combines a staged quinary-to-ternary activation quantizer with a channel-wise tail-risk loss, which together stabilize training under ternary constraints. We further introduce a quantization-aware frequency router that learns a low–high decomposition and adaptively allocates energy to the appropriate branch. Extensive experiments demonstrate competitive accuracy across diverse visual benchmarks, while achieving substantial model compression, faster inference, and lower energy consumption.

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

# A  APPENDIX

## A.1  LLM USAGE STATEMENT

We used a large language model solely to improve clarity and grammar, and all output was reviewed by the authors.

## A.2  DETAILED QUANTIZATION SETTINGS

For the backward pass we employ a bounded straight-through estimator. Let $g = \frac{\partial L}{\partial y}$ and $u = \bar{x}/\alpha$ define the saturation mask $M = \mathbf{1}(|u| \leq Q)$. Then the gradient with respect to the input is $\frac{\partial \mathcal{L}}{\partial \bar{x}} = g \odot M$ and the gradient with respect to the scale follow from Equation (7) and approximating $\frac{\partial L}{\partial y} \approx M$ yields

$$\frac{\partial y}{\partial \alpha} \approx q_Q(u) - \frac{x}{\alpha} M, \qquad \frac{\partial \mathcal{L}}{\partial \alpha} = \sum_{n \in \mathcal{I}_c} g_n \left( q_Q(u_n) - \frac{x_n}{\alpha_c} M_n \right), \qquad (15)$$

where $\mathcal{I}_c$ indexes all elements of channel c. Following LSQ, we apply a per-channel rescaling

$$\frac{\partial \mathcal{L}}{\partial \alpha_c} \leftarrow \frac{1}{\sqrt{N_c}\, Q} \frac{\partial \mathcal{L}}{\partial \alpha_c}. \qquad (16)$$

with $N_c = |\mathcal{I}_c|$, which balances the magnitudes of scale gradients with that of activation gradients and stabilizes training under changing $Q$. This formulation yields a compact, implementation-friendly quantizer: training uses only a learnable scale and an integer radius $Q$, and inference emits ternary codes amenable to bit-wise matrix multiplication in all linear projections.

## A.3  DETAILED SOLUTIONS FOR BUDGETED CHANNEL SELECTION

At each selection step we must decide which input channels should temporarily retain five-level activations under a global compute budget. For channel $c \in \{1, \ldots, C\}$ we introduce a binary decision $m_c \in \{0, 1\}$ indicating whether this channel remains five-level ($m_c = 1$) or is set to ternary ($m_c = 0$) for the next interval. The value of keeping a channel five-level is quantified by the per-channel benefit $\Delta R_c$ (Section 4.2.1), which measures the predicted drop in task loss when toggling only $Q_c$ from ternary to five-level while holding all other channels fixed; this value is computed efficiently in a single backward pass via our first-order influence score. To keep the selection mechanism simple and fast, we adopt a unit-cost model in which each five-level channel consumes one budget unit, i.e.,

$$\Delta B_c = 1 \quad \text{and} \quad \sum_{c=1}^{C} m_c \leq B_{\max}(t), \qquad (17)$$

where $B_{\max}(t) \in \mathbb{N}$ denotes the maximum number of five-level channels allowed at time $t$. The selection problem is thus a 0–1 knapsack:

$$\max_{m_1, \ldots, m_C} \sum_{c=1}^{C} m_c \, \Delta R_c \quad \text{s.t.} \quad \sum_{c=1}^{C} m_c \leq B_{\max}(t), \ \ m_c \in \{0, 1\}. \qquad (18)$$

With identical costs this reduces to choosing the top-$B_{\max}(t)$ channels by $\Delta R_c$:

$$\mathcal{S}_t = \underset{|\mathcal{S}|=B_{\max}(t)}{\arg \text{top}} \ \{\Delta R_c\}_{c=1}^{C}, \qquad (19)$$

after which we set $m_c = 1$ for $c \in \mathcal{S}_t$ and $m_c = 0$ otherwise. Computing $\{\Delta R_c\}$ requires no extra forward passes beyond the standard training step, and the selection itself costs $O(C \log C)$ for sorting (or $O(C)$ with partial selection). To prevent churn and encourage stable kernels, we update the mask every $T$ steps, impose a minimum dwell time so that selected channels remain five-level for at least $D$ rounds before being reconsidered, and decay the budget $B_{\max}(t)$ over epochs to implement a smooth curriculum from five-level to ternary. For kernel-friendly deployment we may optionally operate on non-overlapping channel groups (e.g., size $G$) and apply the same unit-cost rule at the group level by ranking groups using the sum of their member benefits.

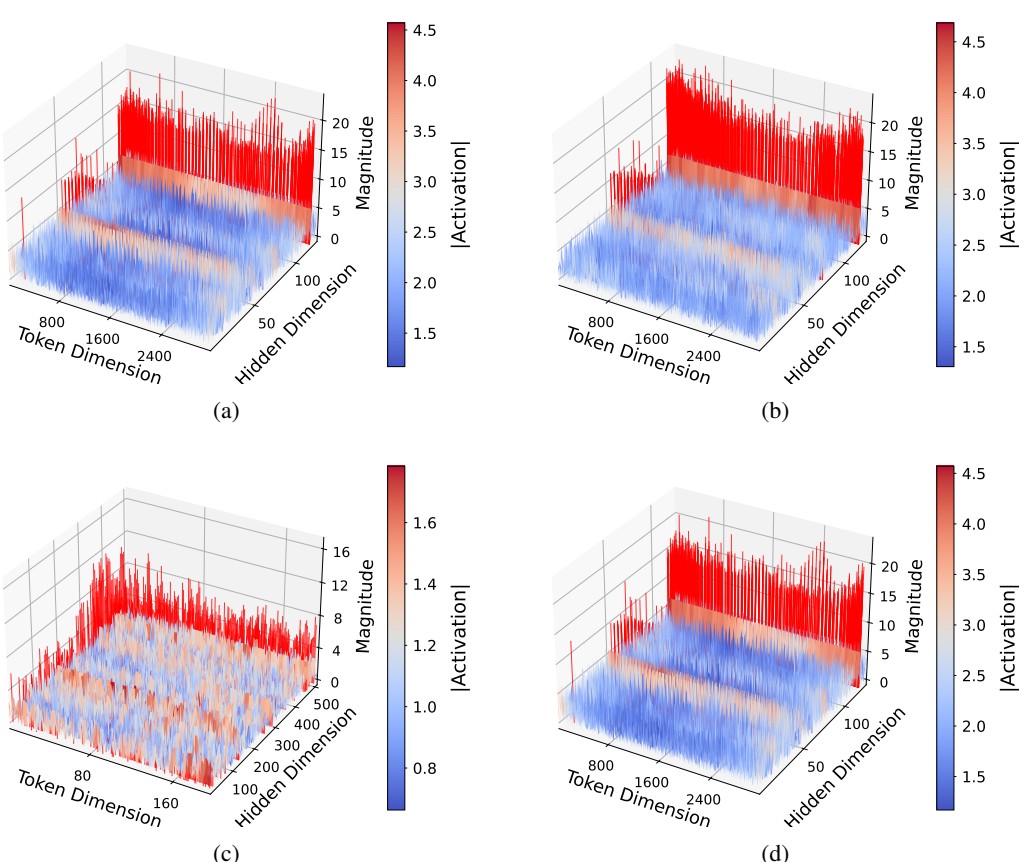

Figure 4: 3D activation landscapes on four inputs activations of In_Proj layers.

## A.4    ADDITIONAL ACTIVATION VISUALIZATION

We provide additional diagnostics to complement the main text. Figure 4 visualizes the activation landscape of the In_Proj layer on four representative inputs. Across inputs we consistently observe tall, sparse spikes concentrated in a small subset of channels, while the majority of channels remain in a low-amplitude regime. This pattern confirms that the channel-wise distribution is highly skewed, and that the outlier channels are not idiosyncratic to a single sample but recur across images.

Figure 5 examines ternary code occupancy at different layers and epochs. For each row, the left bar charts report the occupancy of five randomly sampled channels, revealing strong per-channel heterogeneity—some channels spend most of the time at $\pm\alpha$. whereas others remain near zero. The right violin plots aggregate over all channels and show how the global distribution evolves with training: the mass near zero grows when ternarization compresses the dynamic range, while deeper layers exhibit heavier tails at $\pm\alpha$. Together, the figures substantiate two claims made in the paper: activation statistics are dominated by a few hot channels, and naive ternarization amplifies inter-channel imbalance over time. These observations motivate our staged five-to-three activation quantizer and the tail-risk regularization that stabilize scale estimation and reduce collapse in later layers.

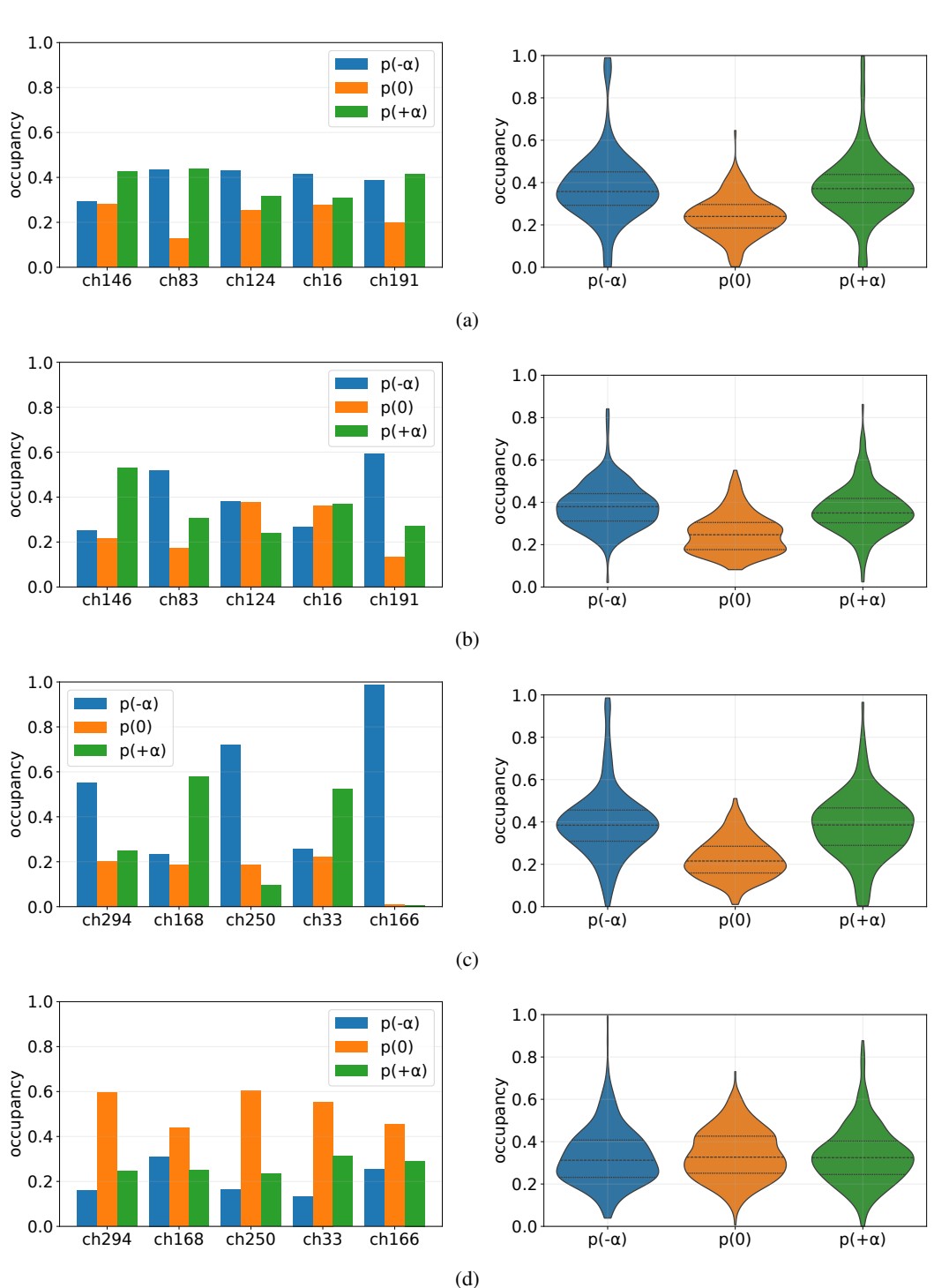

Figure 5: Per-channel occupancy diagnostics at different layers and epochs. For each row: left shows five randomly sampled channels; right shows the violin plot over all channels.

