# OpenReview forum: "TVMamba:Towards Efficient Visual Mammba With Ternary Weights and Activations"
_ICLR.cc/2026/Conference — ICLR 2026 Conference Withdrawn Submission_

### Official Review · Reviewer_ySrc · 2025-10-15

**Soundness:** 3
**Presentation:** 2
**Contribution:** 2
**Rating:** 2
**Confidence:** 4

**Summary:**

This paper proposes TVMamba, a framework for full ternary quantization (ternary weights + activations) of Visual Mamba to boost deployment efficiency on resource-constrained devices. It addresses ternary activation’s key issues—optimization instability and spectral distortion—via customized designs: Q2T (quinary-to-ternary staged quantization) with CTRL loss, and QAFR (quantization-aware frequency routing) with L_QHF loss. Experiments across ImageNet-1K classification, MSCOCO detection/segmentation show a balance between accuracy (small FP16 gap) and efficiency (storage compression, MatMul speedup), but the work has flaws: confusing writing, missing experimental details (e.g., QAT time, inference speed), insufficient theoretical derivation, and limited loss function innovation (mostly engineering tweaks).

**Strengths:**

1. Fills the gap in the field of "full ternary quantization" for Visual Mamba.
2. Addresses the two unique pain points of ternary activation (optimization instability and spectral distortion) with tailored designs, solving key issues that existing methods struggle to adapt to.
3. The experimental design is not limited to a single task or metric. Instead, it verifies the solution’s value from two aspects—"task generality" and "efficiency comprehensiveness"—avoiding concerns about "single-task overfitting".

**Weaknesses:**

1. **Problems in Manuscript Writing (Logical Confusion, Imprecise Expression)**:
   (1). The core analysis of "why ternary quantization is difficult" (optimization instability, spectral distortion) is embedded in Section 4.1 of "4. Method Design", rather than being placed in the "Introduction" or a separate "Problem Definition" section. When readers encounter solutions like "Q2T staged quantization" and "QAFR frequency routing", they have not yet clarified "what specific problems these solutions aim to solve", resulting in a logical inversion of "seeing the solution before understanding the problem".
   (2). After explaining the Selective Scan principle of SSM in Chapter 3 "Preliminaries", Chapter 4 directly jumps to "4.2.1 Q2T Design" without explaining the connection between "SSM characteristics in the preliminaries and subsequent quantization solutions" (e.g., how the low-pass characteristic of SSM causes spectral distortion and requires QAFR for resolution).
   (3). For the QAFR routing weight formula in Section 4.2.2:
   $$[g_{low}, g_{high}] = \text{Softmax}(\varphi(E_{low}, E_{high}, \alpha, \text{sat}))$$
   The calculation method of "$E_{low}/E_{high}$ (channel-wise spectral-energy descriptors)" is not defined. As a result, the formula becomes a mere "symbol pile-up".

2. Critical metrics such as the time required for Quantization-Aware Training (QAT) and the actual inference speedup of the network for a single image are not reported.

3. Separate ablation validations for the three components (Q2T, QAFR, Q-HF) are not conducted.

4. Key parameters such as the k-value of CTRL, the initial value and decay strategy of $B_{max}(t)$, and the kernel size of QAFR operators are not provided.

5. There is no explanation for why only the low-pass characteristic of SSM is amplified—should the high-pass characteristic be more amplified after SSM’s PScan?

6. Missing Mathematical Derivation for Loss Functions. The mathematical principles of CTRL/L_QHF (e.g., the formula for the impact of outlier errors on gradients, the quantitative relationship between high-frequency errors and spectral distortion) are not derived. The effectiveness is only inferred from experimental results, making these designs "empirical" in nature.

7. Limited Innovation in Core Methods. The first method (quinary quantization followed by ternary quantization) is a common practice in engineering. The main contribution lies in loss function improvements, which are more like engineering application optimizations rather than fundamental innovations.

8. CTRL essentially narrows "global regularization" to "Top-k% samples within a channel", which is a minor adjustment of "scope reduction" rather than a "mechanistic innovation". For example, if the k-value of CTRL is set to 100%, it degrades to per-channel MSE loss, which is identical to existing methods.

9. The core logic of L_QHF—"using a high-pass filter to extract high-frequency errors between the quantized model and the full-precision model, and aligning these errors through distillation to mitigate spectral distortion"—is essentially a "migration of 'spectral loss' from super-resolution/image restoration tasks to quantization tasks".

10. No comparative experiments are conducted with existing loss functions (per-channel MSE, L1 regularization, general spectral loss), making it impossible to prove that "the performance of CTRL/L_QHF is significantly superior to existing methods". Thus, the "irreplaceability" of the innovation is weak.

Based on the above issues, I will give a rejection score. I look forward to the authors addressing the above questions in the revised manuscript, and I am happy to reconsider and raise my score.

**Questions:**

See "Weaknesses"

---

### Official Review · Reviewer_qmdd · 2025-10-19

**Soundness:** 3
**Presentation:** 3
**Contribution:** 3
**Rating:** 4
**Confidence:** 4

**Summary:**

This paper introduces TVMamba, a framework that jointly ternarizes both weights and activations (W3A3) in Visual Mamba backbones. The authors identify that naive ternarization leads to unstable optimization and severe spectral distortion due to uneven activation distributions across channels. To mitigate this, TVMamba employs a quinary-to-ternary (Q2T) activation quantizer that gradually anneals activations from five levels to ternary, assisted by a Channel-wise Tail-Risk Loss (CTRL) to handle outliers. Furthermore, a Quantization-Aware Frequency Routing (QAFR) module is proposed to preserve high-frequency visual details lost under low-bit constraints. Experiments on VMamba and Vim show that TVMamba maintains competitive ImageNet, COCO, and segmentation accuracy with over 4× model compression, up to 86.6× MatMul speedup, and substantial energy savings. Overall, TVMamba achieves state-of-the-art ternary quantization for state-space vision models with minimal accuracy loss.

**Strengths:**

The paper presents a full ternary quantization framework (W3A3) for Visual Mamba models. To address activation imbalance and frequency-domain distortions, the authors propose principled solutions: the Quinary-to-Ternary (Q2T) activation quantizer and the Quantization-Aware Frequency Routing (QAFR) module. The experimental evaluations are comprehensive, covering multiple downstream tasks such as image classification and segmentation, and show consistent improvements over state-of-the-art baselines. Furthermore, the paper demonstrates hardware-level efficiency through detailed micro-benchmarks, highlighting the potential for end-to-end acceleration in real-world deployment. Overall, the work is technically sound, well-structured, and makes a meaningful contribution to efficient quantization of state-space vision models.

**Weaknesses:**

### Major concerns
- The latency profiling presented through the micro-benchmark (i.e., matrix multiplication) appears questionable. To the best of my knowledge, BitNet (W3A16) [1], which employs ternary weight-only quantization, reports 1.37×–5.07× speedups on ARM CPUs (versus FP16). In contrast, the reported 17×–87× acceleration in this paper seems unrealistically high without further clarification of experimental setup, kernel implementation, or hardware configurations.
- While the paper includes micro-benchmark results to illustrate potential computational savings, it lacks end-to-end latency profiling on full model inference or real deployment scenarios. Such results are essential to validate that the observed kernel-level speedups translate into actual runtime gains in practical settings.
- It would further strengthen the paper to include comparisons with different quantization precisions, such as W4A4 and W4A16, to better situate the proposed method within the broader efficiency–accuracy trade-off landscape. Presenting a Pareto frontier that contrasts the baseline models and the proposed W3A3 approach would provide clearer insight into the achievable balance between compression ratio, accuracy, and latency. Moreover, such analysis could highlight practical deployment trade-offs — for instance, W3A3 may achieve superior throughput in cloud-scale serving with large batch sizes, whereas W3A16 or W4A16 configurations could offer better accuracy for personal or edge inference where batch size is small (e.g., batch = 1).

### Minor
- The section of quantization-aware frequency routing is hard to follow. I would recommend to add a figure to explain on the mechanism.


[1] BitNet (https://github.com/microsoft/BitNet/tree/main)

**Questions:**

- Could the authors elaborate on how the 86.6× speedup reported in Table 6 was achieved? The magnitude of improvement appears unusually high compared to prior ternary quantization studies (e.g., BitNet [1]) and would benefit from clarification regarding the benchmark setup, kernel implementation, and measurement methodology.
- Could the authors provide end-to-end latency profiling on either CPUs or GPUs? Kernel-level benchmarks are informative, but demonstrating actual inference-time gains would more convincingly support the efficiency claims.
- It would also be helpful if the authors included W3A16/W4A16/W4A4 latency results as an additional baseline in Table 6. This comparison would help contextualize the benefit of full ternary quantization relative to mixed-precision schemes that preserve activation precision.
- Finally, the paper could be further strengthened by comparing across different quantization precisions (e.g., W4A4, W4A16) and illustrating a Pareto frontier that captures the trade-off between accuracy, latency, and model size. Such an analysis would more comprehensively showcase where the proposed W3A3 configuration lies in the efficiency–accuracy spectrum and provide actionable guidance for various deployment scenarios (e.g., cloud serving vs. on-device inference).

I am willing to adjust my rating if the above concerns are addressed by the follow-up discussions.

---

### Official Review · Reviewer_6Tui · 2025-10-31

**Soundness:** 3
**Presentation:** 3
**Contribution:** 3
**Rating:** 4
**Confidence:** 5

**Summary:**

1. This paper presents TVMamba, the first to ternarize both weights and activations in Visual Mamba, boosting efficiency for resource-constrained devices .
2. It solves ternary quantization issues (unstable optimization, spectral distortions) via a quinary-to-ternary codebook and a quantization-aware frequency router (QAFR) .
3. The codebook uses five-level activations for training (with CTRL loss) and collapses to ternary for deployment .
4. QAFR splits features into low/high frequencies, routing them to SSM core and an enhancer, with minimal overhead .
5. Experiments on VMamba/Vim show TVMamba-T has ~3.2-3.5% FP16 accuracy gap, 3.55-4.35× storage compression, and 17×–87× MatMul speedup .
6. Ablations confirm the quinary-to-ternary quantizer drives main accuracy gains (~14% avg) .

**Strengths:**

1. It innovatively realizes the joint ternarization of weights and activation values for the visual Mamba model. As the first framework of its kind in this field, it breaks through the 8-bit activation limitation.
2. The designed quinary-ternary progressive quantization and quantization-aware frequency routing module effectively address the issues of optimization instability and high-frequency detail loss caused by ternary quantization, achieving a good balance between efficiency and accuracy.
3. The charts and graphs in the paper are very clear and can effectively support many of the author's arguments.

**Weaknesses:**

1. It is a consensus that the weight parameters of linear layers are the main part of weight quantization, so the presentation in Table 1 is rather redundant.
2. The PPL and Zero-shot experiments are missing, which are important metrics for measuring the effectiveness of quantization work.
3. The quantized models are limited and have not been extended to more SSM architectures.

**Questions:**

The problem that ternary quantization struggles to handle outliers also exists in the Transformer architecture, so why can't its solutions be directly adopted?

---

### Official Review · Reviewer_tc4g · 2025-11-02

**Soundness:** 2
**Presentation:** 1
**Contribution:** 2
**Rating:** 2
**Confidence:** 2

**Summary:**

The paper proposes a two-stage quantization method. Firstly, the activations are quantized to quinary, and secondly, they are quantized to ternary using a step-wise approach.

**Strengths:**

The quantization of the activations from 8 bits to ternary is novel and significant.
Section 4.2.1 is presented quite well.

**Weaknesses:**

1. There are several grammatical errors.
2. There are several acronyms that are not defined (e.g., CLS and KLT).
3. The paper should be self-contained. It should not be presumed that the target audience is experts in the particular topic of the paper. Towards this, a way more dedicated and detailed section for preliminaries is required.
4. The proposed methodology is presented at a too high level of abstraction. A more detailed and rigorous mathematical formulation is required.
5. AP metric is not defined.
6. Tables 5 and 6 do not appear in the same order they are referred to.

**Questions:**

The preliminaries are presented quite poorly. What is h-dot in (1)? Why is the discrete form of B different from that of A? From (1) and (2), it is evident that y is a function of D. However, why is there no D in (3)? What does "|," denote in (4)?

Section 4.2.2 is presented quite poorly. What is the depthwise separable smoothing operator in (11)? What is the fixed high-pass operator in (14)?

---

### Note · Authors · 2025-11-12

I have read and agree with the venue's withdrawal policy on behalf of myself and my co-authors.